# Coherent control of asymmetric spintronic terahertz emission from two-dimensional hybrid metal halides

Kankan Cong[1,12], Eric Vetter[2,3,12], Liang Yan [ORCID] [3,4,12], Yi Li[5,6], Qi Zhang[1,11], Yuzan Xiong[5,7], Hongwei Qu[7], Richard D. Schaller [ORCID] [8], Axel Hoffmann[6,9], Alexander F. Kemper [ORCID] [2], Yongxin Yao [ORCID] [10], Jigang Wang[10], Wei You [ORCID] [3,4 ✉], Haidan Wen [ORCID] [1 ✉], Wei Zhang[5,6 ✉] & Dali Sun [ORCID] [2,3 ✉]

Next-generation terahertz (THz) sources demand lightweight, low-cost, defect-tolerant, and robust components with synergistic, tunable capabilities. However, a paucity of materials systems simultaneously possessing these desirable attributes and functionalities has made device realization difficult. Here we report the observation of asymmetric spintronic-THz radiation in Two-Dimensional Hybrid Metal Halides (2D-HMH) interfaced with a ferromagnetic metal, produced by ultrafast spin current under femtosecond laser excitation. The generated THz radiation exhibits an asymmetric intensity toward forward and backward emission direction whose directionality can be mutually controlled by the direction of applied magnetic field and linear polarization of the laser pulse. Our work demonstrates the capability for the coherent control of THz emission from 2D-HMHs, enabling their promising applications on the ultrafast timescale as solution-processed material candidates for future THz emitters.

[1] Advanced Photon Source, Argonne National Laboratory, Argonne, IL 60439, USA. [2] Department of Physics, North Carolina State University, Raleigh, NC 27695, USA. [3] Organic and Carbon Electronics Lab (ORaCEL), North Carolina State University, Raleigh, NC 27695, USA. [4] Department of Chemistry, University of North Carolina at Chapel Hill, Chapel Hill, NC 27599, USA. [5] Department of Physics, Oakland University, Rochester, MI 48309, USA. [6] Materials Science Division, Argonne National Laboratory, Argonne, IL 60439, USA. [7] Department of Electronic and Computer Engineering, Oakland University, Rochester, MI 48309, USA. [8] Center for Nanoscale Materials, Argonne National Laboratory, Argonne, IL 64039, USA. [9] Department of Materials Science and Engineering, University of Illinois at Urbana-Champaign, Urbana, IL 61801, USA. [10] Ames Laboratory and Department of Physics and Astronomy, Iowa State University, Ames, IA 50011, USA. [11] Present address: Department of Physics, Nanjing University, 210093 Nanjing, P. R. China. [12] These authors contributed equally: Kankan Cong, Eric Vetter, Liang Yan. ✉email: wyou@unc.edu; wen@anl.gov; weizhang@oakland.edu; dsun4@ncsu.edu

Terahertz (THz) technologies hold great promise for the development of advanced imaging, sensing, security, and communication applications[1–3]. Spintronic-THz, an alternative route for producing THz radiation using ultrafast spintronics, has recently emerged as a prominent field in conjunction with magnetism, photonics, and ultrafast electro-optics[1]. It holds vast technological advantages that potentially outperform current THz emitters based on nonlinear optics in terms of ultrabroad bandwidth (e.g., 1–30 THz)[4,5], ultrafast phase control[6], light-wave acceleration of long-lived currents[7,8], scalability, and cost[2]. The seminal works on spintronic-THz emitters using ferromagnet (FM)/nonmagnetic metallic heterostructures, which convert laser-generated, ultrafast spin current bursts into THz pulses via the bulk inverse spin Hall effect (ISHE)[9] (Fig. 1a), helped drive the exploration of THz generation at Rashba interfaces between two nonmagnetic materials (e.g., two-dimensional electron gas, 2DEG) via the inverse Rashba-Edelstein effect (IREE)[10–12]. The ultrafast electric current, $\mathbf{J}_c$, that produces the transient THz emission carried by an interfacial state can be described by:

$$\mathbf{J}_c \propto \lambda_{\mathrm{IREE}} \mathbf{J}_s \times \mathbf{s} \qquad (1)$$

where $\lambda_{\mathrm{IREE}}$ is the spin-to-charge conversion coefficient, also known as the IREE length in the 2D limit[12], $\mathbf{s}$ is the direction of the injected spin polarization parallel to the Rashba interface, and $\mathbf{J}_s$ is the femtosecond laser-generated superdiffusive spin current from the adjacent ferromagnetic material (see Fig. 1a). Since an efficient spin-to-charge conversion benefits from the 2D limit of the cross-section at a Rashba state[13], the recent demonstration of spintronic-THz emitters using the Bi/Ag 2DEG motivates materials research to harness interfacial Rashba systems as efficient spintronic-THz sources[5,6].

Two-Dimensional Hybrid Metal Halides (2D-HMHs), a rising star in synthetic semiconductors prepared by cost-effective, low-temperature solution processing, have recently attracted much research interest[14–16]. 2D-HMHs have been shown to allow for facile and economical, solution-based crude synthesis while still maintaining high energy conversion efficiency, versatile chemical flexibility, quantum well and dielectric confinement effects, and exceptional defect tolerance[17]. These traits have initiated a myriad of interdisciplinary device applications in 2D-HMHs beyond their revolutionary success in photovoltaics[18]. However, their rich spintronic functionalities have yet to be utilized, despite

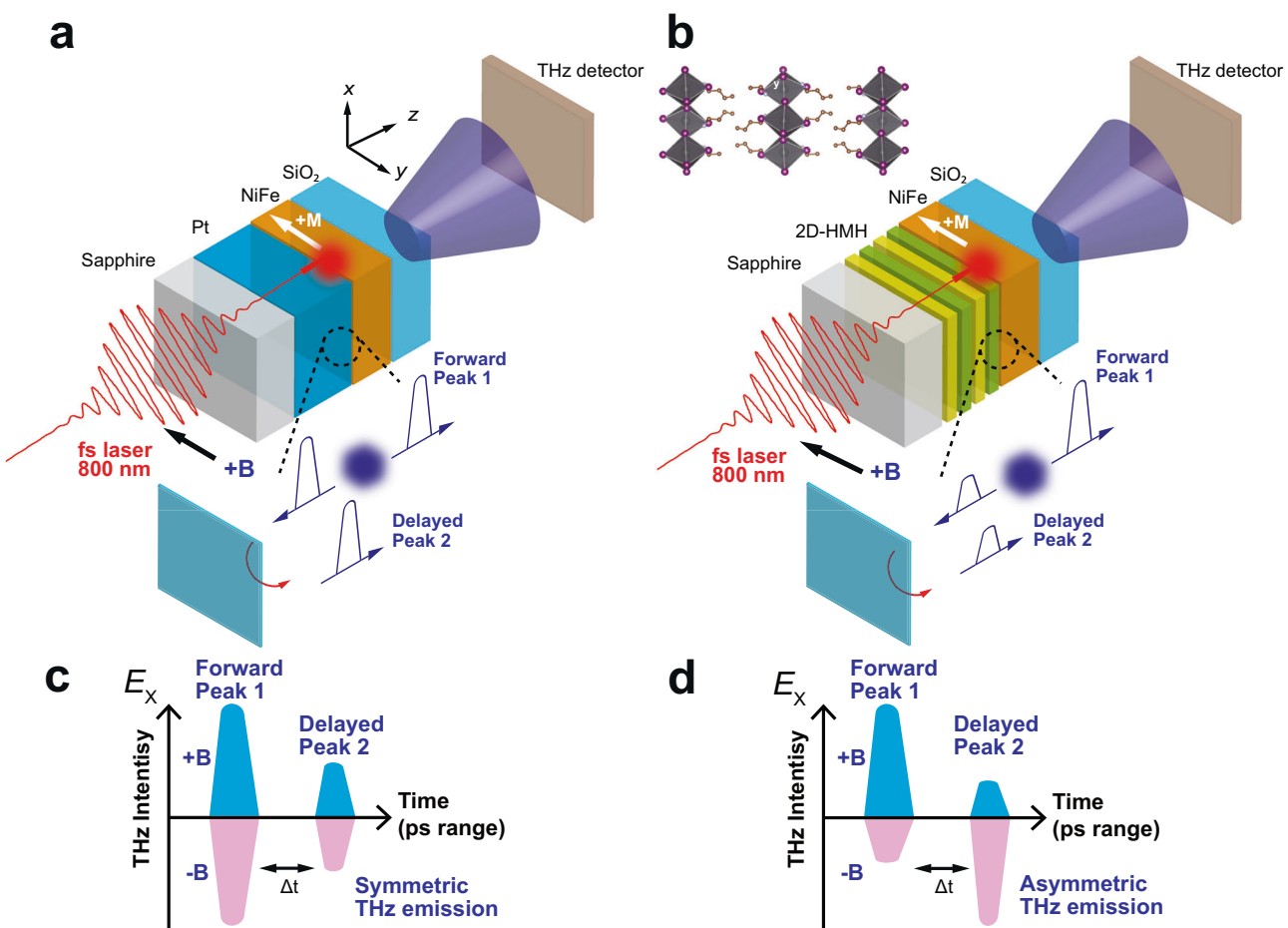

**Fig. 1 Schematic illustration of spintronic-THz emission in Pt/NiFe and 2D-HMH/NiFe heterostructures. a** A sketch of metallic-based spintronic-THz emission in a Sapphire/Pt/NiFe heterostructure. Under fs laser excitation, the superdiffusive spin current injects into the nonmagnetic metallic layer (e.g., Pt) and produces substantial THz radiation via an efficient spin-to-charge conversion process. Generated THz radiation propagates toward the forward ($+z$, i.e., forward peak 1) and backward ($-z$) direction evenly. The backward THz radiation is reflected at the interface between the sapphire substrate and air, generating a delayed THz signal (delayed peak 2) collected by the THz detector ($\Delta t = t_1 - t_2 \sim$10-11 picoseconds). **b** Schematic structure of the hybrid spintronic-THz emitter using layered 2D-HMH materials and magnetic field-dependent THz radiation from the metallic spintronic-THz emitters along the x-direction in the picosecond timescale. The phase of THz emission depends on the magnetization of the NiFe. **c, d** The illustrated symmetric and asymmetric THz radiation under fs laser excitation, respectively. An asymmetric THz radiation emits toward $+z$ and $-z$ direction, of which the intensity is controlled by the applied external magnetic field. The top-left panel in **b** shows the schematic crystal structure of prototypical 2D-HMH materials, i.e., two-dimensional $(BA)_2PbI_4$.

theoretical predictions and a handful of pioneering investigations on their large spin-orbit interaction and emergent Rashba state[19–22]. The 2D layered HMHs naturally form 'quantum well-like' structures consisting of a self-assembled periodic array of inorganic perovskite layers, constituting corner sharing $PbI_6$ octahedral slabs (acting as 'wells') separated by organic spacers (acting as 'barriers') in their lattice framework. In contrast to the bulky organic cations composed of only light elements, the periodically layered inorganic $PbI_6$ frameworks dominate the spin-orbit coupling of this material (Fig. 1b, inset), suggesting that layered Rashba states may offer an intriguing spin-to-charge conversion in contrast to that from the single-layer Rashba interface in 2DEGs. The combination of these properties stimulates interest in developing solution-processed, hybrid spin-to-charge transducers on the femtosecond timescale using 2D-HMH materials for a variety of photonic and THz applications[23].

Here we present the report of asymmetric spintronic-THz emission in 2D-HMHs/ferromagnet heterostructures at room temperature. We show that the emitted THz electric fields can be generated by an ultrafast, transient spin current pulse from the thin, adjacent FM layer (NiFe) followed by spin-to-charge conversion at femtosecond timescales in 2D-HMH materials (Fig. 1b). We find that upon reversing the external magnetic field polarity, both the phase and emission intensity of the THz electric field can be coherently controlled (Fig. 1b), in contrast to that in metallic heterostructures and 3D-HMH materials where the emitted THz field intensity is mostly independent of the magnetization direction and laser polarization. Our work shows that 2D-HMHs would be desirable material candidates for spintronic-THz generation and manipulation. 2D-HMH materials may prove superior to current conventional semiconductor materials for THz applications, which require sophisticated deposition approaches that are more susceptible to defects. The improved stability and scalable thin-film process of the reduced-dimensional HMH materials enable us to meet the emerging needs for low-cost THz sources with coherent control capabilities.

## Results

**Hybrid spintronic-THz emitter fabrication.** Figure 1b illustrates the principle of operation and schematic device structure of the hybrid spintronic-THz emitter using 2D-HMH materials. The devices are in the form of thin-film multilayers constituting: transparent sapphire substrate/$BA_2PbI_4$ (BA = butylammonium)/NiFe (5 nm)/$SiO_2$ (50 nm) (see Methods and S.I. Fig. S1). The hybrid THz emitters based on these polycrystalline 2D-HMH thin films are prepared on double-polished sapphire substrates by a low-temperature solution-processed spin coating approach (S.I. Fig. S1). The thickness of the thin film is kept above 100 nm to ensure a smooth and uniform film morphology for the following metal deposition. The quality of the prepared thin film is checked by atomic force microscopy, showing a root mean square (RMS) roughness of ~1.3 nm (S.I. Fig. S2). The layered perovskite crystal structures are validated by X-ray diffraction and are consistent with the literature reports[24]. The optical absorption measurement shows distinctive absorption peaks in the films depending on the quantum well confinement formed between inorganic metal halides and bulky organic ligands[24]. Negligible absorption at 800 nm suggests that the below-gap laser excitation directly interacts with the ferromagnetic layer (i.e., NiFe) without an intensity loss from passing through the 2D-HMH layer in the current device configuration. A flat response of the circular dichroism spectra confirms that there is no indication of chiral structure formation, or negligible preferred circular polarization absorption (S.I. Fig. S3).

Upon linearly polarized fs laser excitation, a superdiffusive spin current is generated in the NiFe layer[25] and, in turn, is injected

into the layered 2D-HMH thin film causing the emission of THz radiation via ultrafast spin-to-charge conversion. The generated THz radiation was recorded through the electro-optical sampling technique[26]. The set of unit vectors $x$, $y$, and $z$ represent the chosen coordinate system. A pair of wire-grid polarizers are used to measure the parallel (i.e., $E_y \perp B_y$) and perpendicular (i.e., $E_x \parallel B_y$) components of THz radiation (see Methods) with respect to the direction of the in-plane magnetic field, $B_y$. $B_y$ was applied by using a reversible two-pole permanent magnet at a constant magnetic field of ~500 Oe, which is adequately large to maintain the magnetization of the NiFe layer along the in-plane direction (i.e., $y$-direction) during THz measurements.

**Magnetic Field dependence of asymmetric THz radiation.** Figure 2a presents the typical measured THz signal $E_x(t)$ in 2D hybrid spintronic-THz emitters at room temperature. Each THz transient is fully inverted when the polarity of the magnetic field is reversed, confirming a spintronic origin: Since the THz signal is produced by the superdiffusive spin current burst generated by the NiFe layer, switching the direction of the magnetization of the NiFe layer reverses the spin polarization of the injected spin current. Consequently, the THz signal polarity (i.e., THz phase) from the 2D-HMH thin film is 180° shifted. The magnetic field-induced phase shift confirms the ultrafast spin-to-charge conversion in 2D-HMH thin films.

Following the THz generation mechanism in the metallic spintronic-THz emitters[3], the THz radiation will emit toward both $-z$ (backward) and $+z$ (forward) directions. Therefore, all measured time traces of the THz signal contain two peak groups. The first one is the forward THz radiation (labeled as 'peak 1' appeared at $t_1 =$ ~5 ps) and the delayed one is the backward THz radiation (labeled as the delayed 'peak 2' appeared at $t_2 =$ ~15 ps) due to the reflection at the interface between the sapphire substrate and the air (Fig. 1b). The time difference ($\Delta t =$ ~10-11 ps) between the two peak groups matches the time delay required for the backward THz pulse to transmit through the 0.5 mm-thick sapphire substrate twice (refractive index: ~3.4). Strikingly, we found the presence of a nontrivial asymmetry in THz intensity in both peak groups depending on the direction of the applied magnetic field. For the first peak group, the intensity of the forward THz radiation at the positive magnetic field (+**B**, blue trace) is roughly two times larger than the one at −**B**. In contrast, the second peak group (i.e., backward THz radiation) exhibits an asymmetry: THz intensity at +**B** is clearly weaker than the one at −**B**. This reversed asymmetry of the THz intensity suggests field-dependent asymmetric THz radiation along $-z$ (backward) and $+z$ (forward) directions (see also the illustration in Fig. 1b).

To verify that the observed asymmetric THz emission is uniquely present in 2D-HMH/NiFe samples, we have conducted spintronic-THz measurements in a series of control samples including NiFe/Pt and bare NiFe films (see Fig. 2b). None of them exhibits a similar asymmetric THz intensity at opposite directions of magnetic fields, ruling out the sample/field misalignment during the THz measurement as a possible source of the observed behavior. Moreover, in all the metallic spintronic-THz devices, the phases of the THz radiation for the 1st and 2nd peak group remain the same except for the decreased intensity that has been attributed to the Fabry-Perot process and multiple reflections between the metallic heterostructure and the substrate[27].

Moreover, the phase of the THz transient in the 2D-HMH/NiFe heterostructures between the 1st and 2nd peak group is clearly inverted, although the time delay ($\Delta t =$ ~10-11 ps) between the two peak groups is similar to those in other metallic THz devices. This distinct feature also helps to eliminate the possibility of THz emission generated by the NiFe itself (which

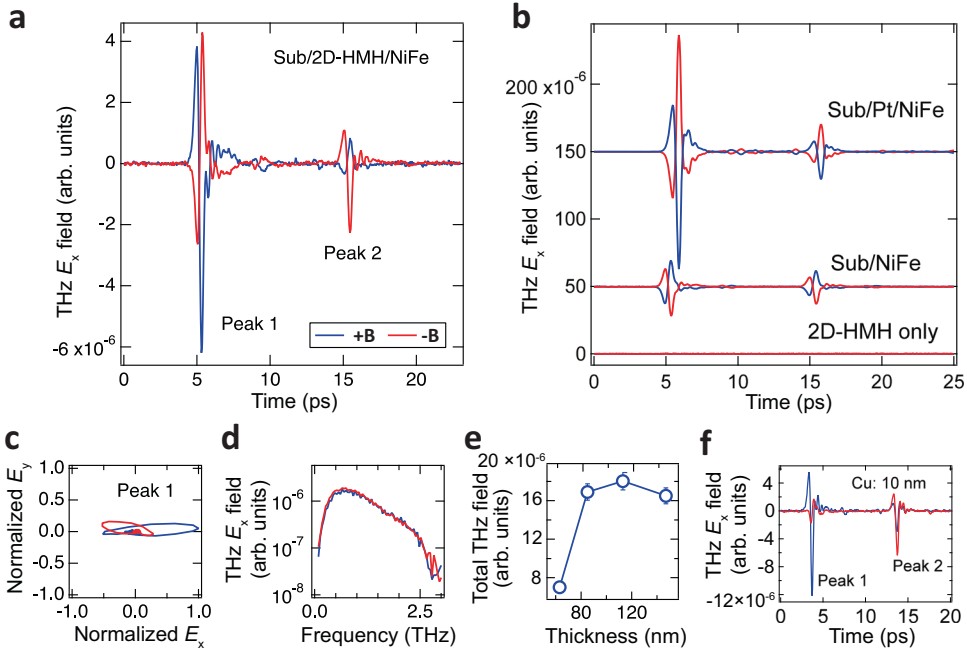

**Fig. 2 Magnetic field-dependent asymmetric THz radiation upon linearly polarized laser excitation. a** Measured time trace of the THz pulse generated from the 2D-HMH/NiFe heterostructures under positive ($+\mathbf{B}$) and negative ($-\mathbf{B}$) magnetic fields, respectively. The forward (peak 1 at $t_1 = \sim5$ ps) and backward (peak 2 at $t_2 = \sim15$ ps) THz pulses exhibit opposite symmetry of the intensity depending on the direction of the applied magnetic field. **b** Symmetric THz radiations are confirmed in NiFe/Pt and NiFe only control samples. No THz signal is observed in the 2D-HMH film without the ferromagnet top-layer under the same laser excitation. **c** The parametric plot of the emitted THz field shows the THz polarization is mainly along the x-direction. **d** The THz emission spectra of **a** indicate equal total THz radiation in spite of asymmetric radiation along with two propagation directions. **e** THz signals show a very weak 2D-HMH thickness dependence at the higher 2D-HMH thicknesses above ~80 nm. The error bars represent the standard errors of the measurements. **f** Pronounced asymmetric THz radiation in 2D-HMH/Cu/NiFe trilayer structure, ruling out the possible proximity effect induced nonreciprocal phenomena at the 2D-HMH/NiFe interface.

should retain identical phases between two peak groups as shown in Fig. 2b), confirming that the dominant mechanism of spintronic-THz generation stems from the 2D-HMH layer via the injected superdiffusive spin current. The observed THz signal is in contrast to the null THz signal measured from the substrate/2D-HMH/SiO$_2$ (see Fig. 2b). Under 800 nm below-gap excitation, we do not observe any measurable THz emission in the 2D-HMH/SiO$_2$ sample within our detection capability, excluding the nonspintronic THz emission due to the photo-Dember effect[28,29], nonlinear optical effect[30], or linear/circular photogalvanic effect which requires photocarriers under the above-gap laser excitation[19]. Magnetic field effect contributing to the THz radiation in the 2D-HMH-only device[31] is unlikely due to the absence of asymmetric THz emission under 400 nm excitation.

A parametric plot of the generated THz electric-field amplitude for the first peak group is presented in Fig. 2c. The forward THz radiation is mainly polarized along the x-direction, perpendicular to the magnetization of the NiFe layer (Fig. 1b), denoting the 'spintronic' nature of THz generation (i.e., $E_x(\text{THz}) \propto \mathbf{J}_s \times S_y$). Figure 2d shows the Fourier spectra of the THz transient signal in the 2D-HMH/NiFe heterostructure for both magnetic field directions. While the 1st and 2nd peak groups exhibit different intensities, the $+\mathbf{B}/-\mathbf{B}$ traces are roughly aligned with each other suggesting an equal intensity of overall THz signals stemming from the same amount of superdiffusive spin current in the 2D-HMH film at opposite magnetic fields. The normalized amplitude spectrum of the THz radiation shows a broader bandwidth compared to the one obtained from the NiFe/Pt (S.I. Fig. S4). The bandwidth goes up to 3 THz, which is partly due to the limitation imposed by the use of a ZnTe crystal in the measurement setup. Figure 2e shows the observed overall THz intensity as a function

of 2D-HMH thickness. The decreased THz signal below 80 nm can be understood as a byproduct of poor thin-film quality that results in decreased spin-to-charge conversion at the interface between NiFe and 2D-HMH layer[16]. The weak thickness dependence of THz intensity at higher thickness implies that the interfacial spin-to-charge conversion is responsible for the spintronic-THz generation rather than the bulk-dominated ISHE process. THz signals have additionally been measured in a trilayer device structure constituting substrate/2D-HMH/Cu (10 nm)/NiFe to further separate the THz signal from the magnetic proximity effect (Fig. 2f). By inserting a thin Cu spacer layer between the 2D-HMH and NiFe layer, the direct proximity effect between the 2D-HMH and NiFe is suppressed, while the superdiffusive spin current can still transmit through the Cu spacer layer and inject into the 2D-HMH layer thanks to the long spin diffusion length of Cu[32]. A pronounced asymmetric THz radiation is still observed when the thickness of the Cu spacer layer is 10 nm. This suggests that the observed asymmetric THz radiations are not attributed to the magnetic proximity effect[33].

**Sample orientation dependence of asymmetric THz radiation.** To further elucidate the origin of asymmetric THz radiation in the 2D-HMH/NiFe heterostructure[34], sample orientation dependence of the THz electric field is measured as shown in Fig. 3. We conducted the THz measurements for two orientations of the sample by flipping the same hybrid heterostructure around the applied magnetic field over 180° so that the incident laser direction was inverted. In Fig. 3a, the laser is incident from the sapphire substrate before successively passing through the 2D-HMH and then the NiFe layers, respectively. Under this condition, the generated spin current was injected back into the 2D-HMH

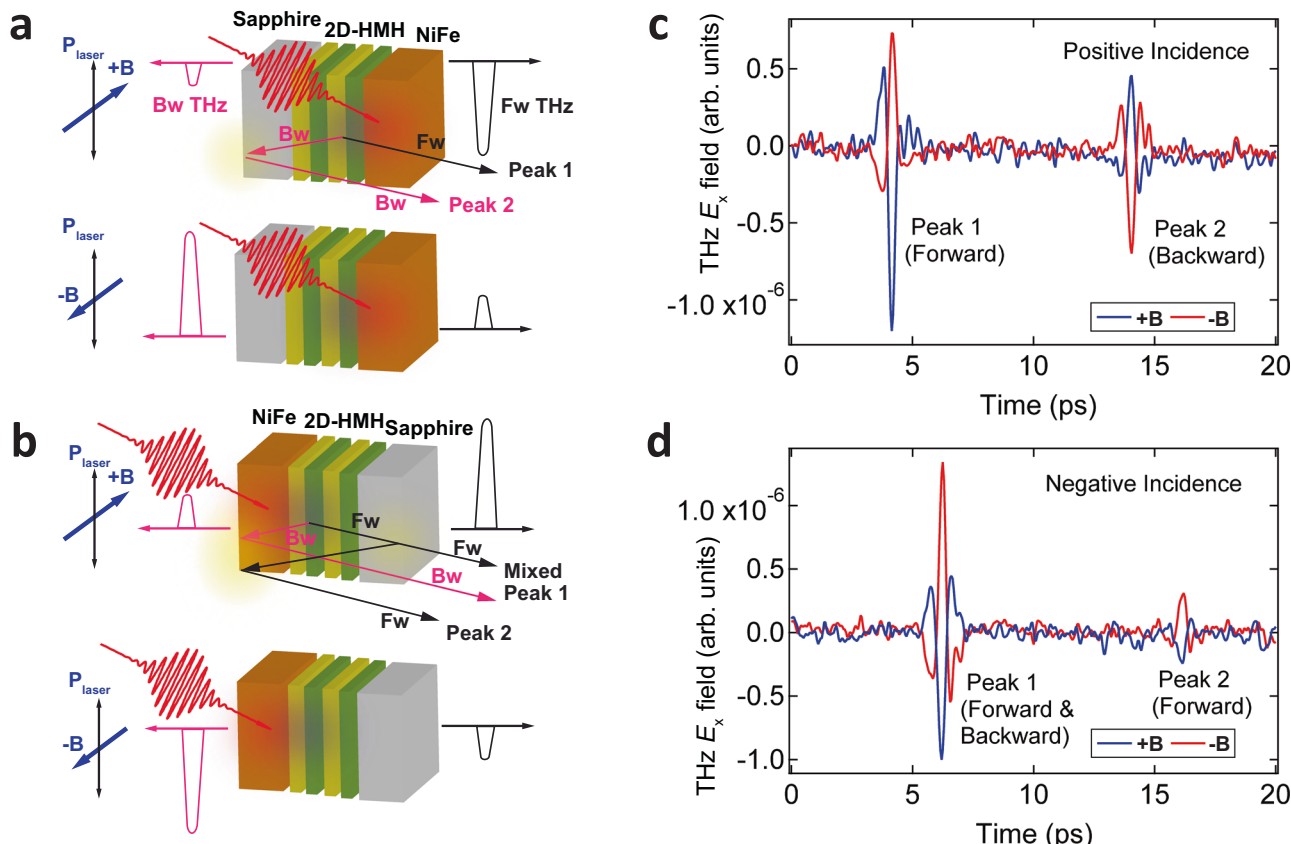

**Fig. 3 Sample orientation dependence of asymmetric THz radiation. a, b** Schematic illustration of the $E_x$ component of generated THz electric field in 2D-HMH/NiFe heterostructure using front and back pump configurations, respectively. The red and blue spots illustrate where the laser pulse is absorbed in the NiFe layer and THz emission is emitted from the 2D-HMH layer, respectively. The black (pink) solid lines represent the optical paths of the forward (backward) THz emission inside the device. The yellow spots indicate the positions where the THz emissions are reflected. **c, d** Obtained electric field of THz radiation along the x-direction as a function of time for opposite incidence direction of light, showing a clear sample orientation dependence of the directionality of the $E_x$ component: The directionality and phase of the $E_x$ component of THz radiation change their polarity with the magnetization and sample flips, respectively.

film opposite to the incident laser direction. The forward THz radiation (peak 1) propagates through the FM layer ($+z$ direction) and the backward peak group ($-z$ direction, peak 2) propagates through the sapphire substrate twice. This configuration is labeled as 'positive incidence' as the normal condition for our THz measurement (Fig. 3a). The 'negative incidence' refers to the situation where the laser was incident to the NiFe layer first. Under this condition, the forward THz radiation (at $t_1 = \sim 6$ ps) will propagate through the sapphire once, shown as the peak 1. Now the delayed peak 2 (at $t_2 = \sim 16$ ps) is actually also originated from the forward THz emission having the same phase as peak 1 because it was reflected at the interface between the air and 0.5 mm-thick sapphire substrate. The backward THz emission under this condition overlaps with the forward THz emission at peak 1 since it cannot be solely separated due to the reflection occurred at the interface between the air and 5 nm-thick NiFe layer (see the illustrated optical paths in Fig. 3a and 3b).

Figure 3c and d clearly shows that the electric THz field changes sign upon the reversal of both sample orientation and magnetic field, respectively. The slight change in delay and timescale of the first THz peak group arises from the different propagation paths of THz radiation and pump light. When the magnetization is fixed (e.g., at $+\mathbf{B}$), turning the sample orientation from the positive to negative incidence geometry does result in an opposite sign of the THz electric field because of the changed diffusion direction of the spin current ($\mathbf{J}_s$). However,

we found the asymmetric THz intensity does not change upon the reversal of sample orientation: THz intensity at $+\mathbf{B}$ is consistently larger than that at $-\mathbf{B}$ regardless of the laser incident direction. Once the angle between the applied magnetic field and linear polarization axis is determined, the asymmetric THz radiation intensity is kept unchanged, although the phase of the THz electric field is fully inverted. This indicates that the role of potential structure-induced spontaneous symmetry breaking at the metallic/hybrid semiconductor interface[34] (a cross product of polar unit vector ($\mathbf{n}_z$) normal to the interface and the applied field, $\mathbf{B}$) is not responsible for the asymmetric spintronic-THz emission, otherwise the weaker intensity of forward THz radiation would be expected in the 'negative incidence' configuration at $+\mathbf{B}$. The lack of spontaneous symmetry breaking in the heterostructure is further corroborated by the absence of reversed $E_y$ component under the circularly polarized light illumination (S.I. Fig. S5)[34].

**Coherent control of THz radiation by pump-light polarization.** As suggested by the sample orientation dependence in Fig. 3, the relative orientation between the laser polarization axis and applied magnetic field may play a key role in determining the asymmetric THz radiation in the hybrid THz emitter. Thus, pump polarization dependence of the THz electric field has been measured below. The geometry of the linearly polarized laser

excitation and THz detection setup is depicted in Fig. 4a and b. Figure 4c and d show the typical 2D contour plot of THz radiation as a function of time and polarization angle, $\Theta$, at the positive and negative magnetic field, respectively. The change of THz intensity in the 1 and 2nd peak group as a function of linear polarization angle in Fig. 4e and f, respectively.

In principle, once the spin polarization and injection direction of the spin current is fixed by the sample structure and magnetization, no dependence of spintronic-THz intensity on the pump polarization should be expected for the spintronic-THz emission as recently reported[4,35]. Strikingly, we found that the emitted THz intensity shows a strong dependence on pump polarization: When applying $+\mathbf{B}$, by rotating the laser polarization axis from 0 to 90 degrees, the amplitude of the forward THz peak group (blue trace in Fig. 4e) keeps decreasing until it reaches a minimum at ~90 degrees while the amplitude of the backward peak group increases (blue traces in Fig. 4f) inversely. At $-\mathbf{B}$, the change of THz amplitude between two peak groups reverse their signs with respect to that at $+\mathbf{B}$. The observed polarization-dependent simultaneous decrease (increase) of THz amplitude in peak 1 and the increase (decrease) in peak 2 rules out the trivial laser-induced absorption or transmission difference for THz pulses in the c-cut sapphire substrate, by which the THz amplitude in both peaks should follow exactly the same trace without any magnetic field control. A large percentage of the THz amplitude is modulated by the pump polarization up to 30–40%, demonstrating the realization of coherent control of THz amplitude in the hybrid spintronic-THz emitter.

Both magnetic field and pump polarization-dependent THz measurements have been performed in 3D-HMH-based hybrid spintronic-THz devices, i.e., $CH_3NH_3PbBr_3$/NiFe. While the 3D $CH_3NH_3PbBr_3$ presumably possesses a similarly large spin-orbit coupling as that in 2D-HMH materials, as well as a surface Rashba state[16] that can produce similar spintronic-THz radiation, neither asymmetric THz radiation nor pump polarization dependence is observed (S.I. Fig. S6). This implies that the observed asymmetric THz radiation strongly correlates with the low-dimensionality of the 2D-HMH layer under below-gap fs laser excitation.

THz transmission measurements have been performed in the same 2D-HMH/NiFe heterostructure without fs laser excitation. In the absence of fs laser excitation, there is no detectable difference for the THz transmission and absorption along both $+z$ and $-z$ directions (S.I. Fig. S7). The THz electric field is also independent of the polycrystalline sample orientation along the in-plane direction, which is corroborated by the magnetic anisotropy measurements, namely rotating the device within the $x$–$y$ plane while keeping the laser pulse linear polarization ($+x$-direction) and the magnetic field ($+y$-direction) constant (S.I. Fig. S8). The measured THz electric-field intensity and polarity shows no difference at different angles, confirming that the asymmetric THz radiation neither originate from the magnetic anisotropy of the NiFe nor the crystalline structure of the 2D-HMH film and its relative orientation with respect to the laser polarization axis.

## Discussion

To put the above asymmetric THz emission findings on a stronger footing, we now consider possible mechanisms that may underpin the observed field- and polarization-dependent asymmetric THz radiation in hybrid spintronic-THz emitters. The control experiments presented in the previous section are inconsistent with several possible causes (see detailed discussion in S.I. section XI) including field-dependent spin-to-charge conversion at the 3D-HMH/NiFe interface (S.I. Fig. S6), THz transmission (S.I. Fig. S7), magnetic anisotropy in the NiFe film (S.I. Fig. S8), and the role of $SiO_2$ capping layer (S.I. Fig. S9-S10).

Thus we extend the IREE mechanism by including an additional in-plane momentum shift of the Rashba bands caused by a joint effect of both the applied magnetic field $B$ and time-dependent electric polarization induced by linearly polarized fs laser pump pulses, as illustrated in Fig. 5a and b. Considering a typical Rashba band splitting[36], being symmetric with respect to spin and momentum[20], we would obtain ultrafast charge current $\mathbf{J}_c$(IREE) follows in $\mp x$ direction if $\mathbf{B}$ is flipped between $\pm y$ direction due to the IREE. Such a transient charge current flow

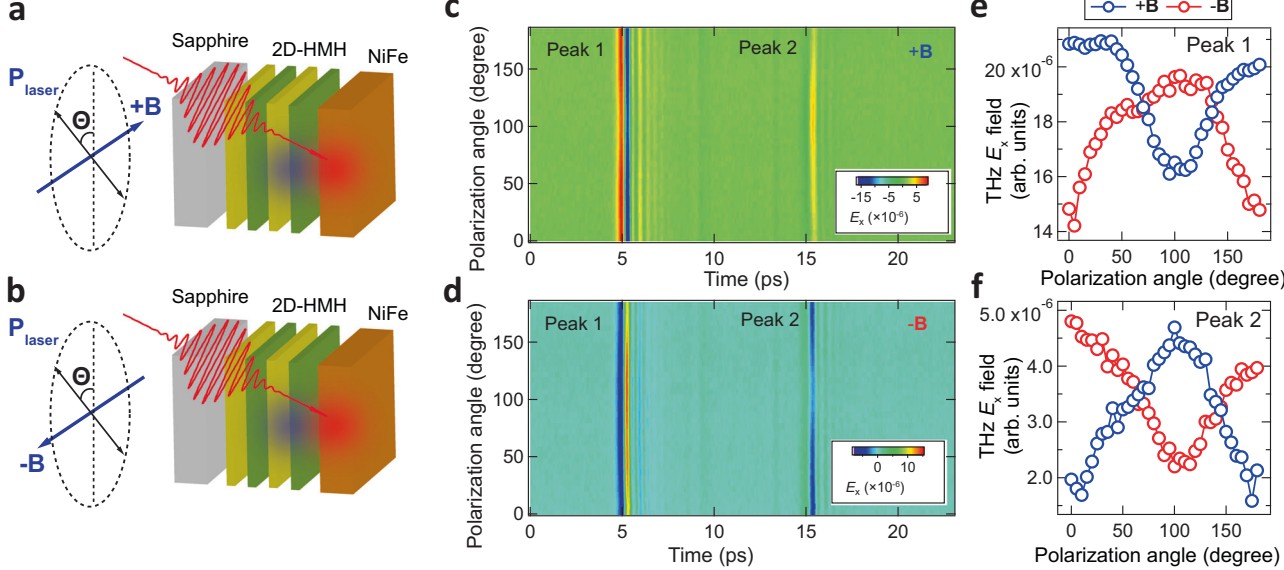

**Fig. 4 Pump polarization dependence of asymmetric THz radiation. a, b** Schematic illustration of pump polarization dependence of THz measurement in 2D-HMH/NiFe heterostructure as a function of the relative angle ($\Theta$) between pump linear polarization axis with respect to the magnetic field. **c, d** 2D contour plot of the electric field of THz radiation along the $x$-direction as a function of time and polarization angle ($\Theta$), exhibiting strong linear polarization-modulated THz intensity in both two peak groups. Panels **e** and **f** summarize the opposite changes of the polarization-dependent THz intensity in Peak 1 and Peak 2 with the switching of the magnetic field, respectively.

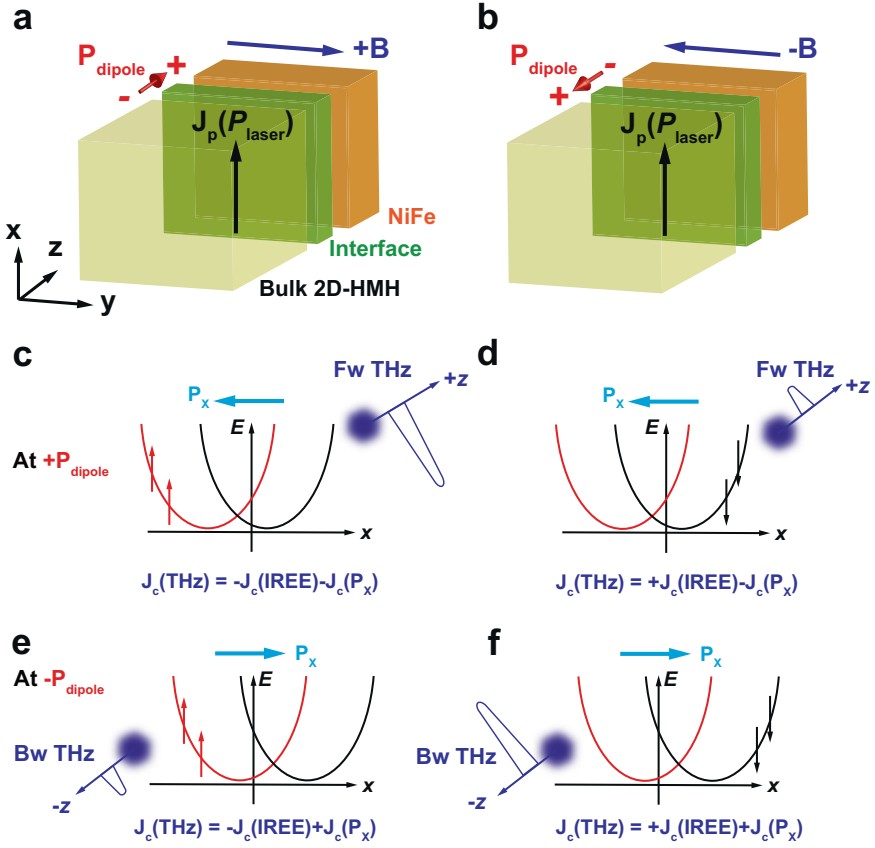

**Fig. 5 Schematic illustration of the shifted Rashba bands and resulting asymmetric THz emission. a, b** Formation of an electric polarization $\mathbf{p}_{dipole}(z)$ along the $z$-axis across the interfacial layered structure of the 2D-HMH film in the vicinity of the 2D-HMH/NiFe interface under positive ($+\mathbf{B}$) and negative ($-\mathbf{B}$) magnetic fields, respectively. $\mathbf{J}_p(P_{laser})$ is the polarization current along the $x$-direction induced by linear polarization of the laser pump pulse. **c, d** A sketch of the shifted Rashba bands in the presence of momentum shift, $\mathbf{P}_x < 0$ at the $+\mathbf{p}_{dipole}(z)$ side according to Eq. (2). The red (black) arrows indicate the injected spin polarization at $+\mathbf{B}$ ($-\mathbf{B}$) magnetic field. **e, f** The shifted Rashba bands at the $-\mathbf{p}_{dipole}(z)$ side ($\mathbf{P}_x > 0$). The generated THz emission containing both the field-dependent IREE and the field-independent momentum shift components is shown at different fields and surfaces that account for the asymmetric THz radiation.

will in turn only generate symmetric THz emission by flipping $\mathbf{B}$, i.e., an equal amplitude with a $\pi$-phase flip similar to that in NiFe/Pt and NiFe only control samples. However, at the 2D-HMH/NiFe interface, a polarization current, $\mathbf{J}_p(P_{laser})$ can be induced by time-dependent laser polarization ($P_{laser}$) of the pump pulse. The $\mathbf{J}_p(P_{laser})$ effect can be significantly amplified at the 2D-HMH, assisted by the Rashba states, i.e., the $\mathbf{J}_p(P_{laser})$ leads to an in-plane momentum shift of the Rashba bands along the $x$-direction, playing a key role in determining the asymmetric contribution to the THz emission amplitude with respect of the applied $\mathbf{B}$ field as shown in Fig. 2a.

A physical picture of shifted Rashba bands is further elaborated below. Primarily, we describe the origin of the in-plane momentum shift of Rashba bands due to the concerted actions of an electric dipole along the $z$-axis and the ordinary Hall effect. Such behavior can be understood as a consequence of the simultaneous breaking of time-reversal and space-inversion symmetry as typically described in two-dimensional quantum well systems[37]:

$$\mathbf{P}_x = \mathbf{P}_{CM} - \frac{\mathbf{B}}{c} \times \mathbf{p}_{dipole}(z) \qquad (2)$$

where the momentum $\mathbf{P}_x$ is shifted from original center of mass motion $\mathbf{P}_{CM}$ by an amount that arises from the Lorentz force acting on an electric dipole along the $z$-axis, $\mathbf{p}_{dipole}(z) = e\mathbf{r}_z$. This leads to a rigid in-plane momentum shift of the Rashba bands

along the $x$-axis perpendicular to the applied external magnetic field.

The formation of the electric dipole could be understood as the interaction between applied magnetic field and light electric field-induced polarization current. The incident pump laser with polarization along the $x$ direction induces a polarization current $\mathbf{J}_p(P_{laser})$, which will keep the same direction when the $\mathbf{B}$ field reverses. In analogy to ordinary Hall effect caused by conduction current, such polarization current simultaneously breaks time-reversal and space-inversion symmetries in the 2D quantum well structure, leading to the electric dipole $\mathbf{p}_{dipole}(z)$ perpendicular to the $\mathbf{B}$ field and light polarization. As illustrated in Fig. 5a and b, the electric polarization $\mathbf{p}_{dipole}(z)$ is formed along the $z$-axis across the interfacial layered structure of the 2D-HMH film in the vicinity of the 2D-HMH/NiFe interface. When the normal direction of the layered 2D-HMH layer is parallel (antiparallel) to the direction of the $k$ vector of the pump pulse, the $+\mathbf{p}_{dipole}(z)$ ($-\mathbf{p}_{dipole}(z)$) is formed in the layered structure that is mostly responsible for the control of the forward (backward) THz radiation.

Whereas the $\mathbf{p}_{dipole}(z)$ will flip its direction when the $\mathbf{B}$ field reverses ($\mathbf{p}_{dipole}(z) \propto \mathbf{J}_p(P_{laser}) \times \mathbf{B}$), the momentum shift $\mathbf{P}_x$ will be along the same direction regardless of the $\mathbf{B}$ field direction according to Eq. (2) ($\propto \mathbf{B} \times \mathbf{p}_{dipole}(z)$), leading to the field-independent momentum shift of the Rashba bands along the $x$-direction as illustrated in Fig. 5c–f. Thus, the recorded THz

emission contains two transient charge currents generated by the conventional IREE process (i.e., $\mathbf{J}_c(\text{IREE})$) and the additional momentum shift, $\mathbf{J}_c(\mathbf{P}_x)$ as described below:

$$\mathbf{J}_c(\text{THz}) = \mathbf{J}_c(\text{IREE}, \mathbf{B}) + \mathbf{J}_c(\mathbf{P}_x, |\mathbf{B}|) \qquad (3)$$

where $\mathbf{J}_c(\text{IREE})$ component is field-dependent that generates symmetric THz amplitude upon the transient spin current injection. Most intriguingly, however, the $\mathbf{J}_c(\mathbf{P}_x)$ component does not change the polarity between two fields, which accounts for the observed asymmetric THz radiation. As for an arbitrary linear polarization, i.e., the laser pulse rotating away from the x-axis, $\mathbf{J}_p(P_{\text{laser}})$ will come from the projection of light polarization along the x-axis, which is consistent with the pump polarization dependence with respect to the applied $\mathbf{B}$ field as shown in Fig. 4.

**Summary**. Our work shows the realization of ultrafast spin-to-charge conversion in 2D-HMH films with THz frequencies. It offers promising routes towards interconversion between photons, charges, and spin states using scalable, printable, solution-processed hybrid compounds. Taking advantage of layered structures in the presence of the injected superdiffusive spin-polarized current, the phase and intensity of THz radiation in 2D-HMH materials can be manipulated by the magnetization state of the adjacent ferromagnet and the pump polarization, offering contactless coherent control of the THz radiation utilizing spintronic toolkits. Our work will launch a promising testbed for designing a wide variety of low-dimensional HMH materials for future solution-based hybrid spintronic-THz emitter applications, bridging the gap between optoelectronics in HMH materials and THz spintronics.

## Methods

**Device fabrication and characterization**. 2D-HMH films were spin-coated on sapphire substrates in a glove box designed for thin-film preparation. The quality of the prepared 2D-HMH films was characterized by XRD and absorption spectroscopy (S.I. Fig. S2), respectively. The surface roughness of films was characterized by AFM (S.I. Fig. S2). Following the spin coating procedure, the prepared 2D-HMH thin films were immediately transferred into a glove-box-integrated deposition chamber without exposure to air. $Ni_{81}Fe_{19}$ and $SiO_2$ layers were then deposited at room temperature using e-beam evaporation with a chamber base pressure of $1 \times 10^{-7}$ Torr and evaporation rates of 0.5 and 2.0 A/s, respectively. The NiFe layer was deposited on a $4 \times 4$ mm$^2$ area of the substrate, while the SiO$_2$ layer was then coated over the entire $5 \times 5$ mm$^2$ substrate as a capping layer to protect the 2D-HMH thin film during the THz measurement. Our in-situ device fabrication is the key to successful hybrid THz measurement, which significantly protects the device from being oxidized or degraded during the preparation stage due to potential oxygen and humidity. The control devices, such as Sapphire/NiFe/Pt, Sapphire/Pt/NiFe, Sapphire/2D-HMH/SiO$_2$, and Sapphire/NiFe/SiO$_2$ were prepared following the same protocol.

**THz emission measurements**. A THz time-domain emission spectroscopy system in transmission geometry is utilized to study the THz emission from RD-HMHs system at room temperature. A Ti: Sapphire laser system with 800 nm center wavelength, 2 kHz repetition rate, and 35 fs pulse duration is used as the laser source. The laser beam is split into a pump beam and a probe beam via a beam splitter. The pump beam with a pump fluence of ~1.0 mJ/cm$^2$ (see S.I. Fig. S11) is used to normally excite the device, from which the emitted THz radiation is collected, collimated, and focused onto a ZnTe crystal (300 μm-thickness, 110-cut) by a pair of off-axis parabolic mirrors. A pair of wire-grid THz polarizers in the frequency range of 0.1–3 THz is used in between the pair of off-axis parabolic mirrors to analyze the THz polarization. The weak probe beam is focused on the ZnTe crystal, overlapping with the generated THz pulses spatially and temporally. The transmitted probe beam passes through a quarter-wave plate and a Wollaston prism and then is detected by a balanced photodetector.

The thickness of the used sapphire substrate (C-plane (0001), MTI Corp.) is 0.5 mm. The reported dielectric constant for this substrate is 11.58 along the c-axis, corresponding to a refractive index ($n$) of ~3.4. The emitted THz pulse is reflected at the interface between the sapphire substrate and the air and passes the substrate twice, which results in a delayed THz peak in the detected THz waveform The total delayed time $\Delta t$ can be calculated by $(0.5 \text{mm} \times 2)/(c/n) \approx 11$ ps, where $c$ is the speed of light in vacuum. The calculated delay time is consistent with our observations.

## Data availability

All data are available in the main text or supplementary materials. Additional data related to this paper may be requested from the authors.

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

## Acknowledgements

E.V. and D.S. acknowledge supports from US National Science Foundation, ECCS-1933297. W.Z. acknowledges partial supports from Michigan Space Grant Consortium, Air-Force Office of Scientific Research under award no. FA9550-19-1-0254, and US National Science Foundation, ECCS-1933301. L.Y. and W.Y. acknowledge supports from US National Science Foundation, ECCS-1933324, and the Center for Hybrid Organic Inorganic Semiconductors for Energy (CHOISE), an Energy Frontier Research Center funded by the U.S. Department of Energy, Office of Science, Office of Basic Energy Sciences. K.C. and H.W. at Argonne are supported by the U.S. Department of Energy, Office of Science, Materials Science Engineering Division, as well as under contract no. DE-SC0012509. Use of the Center for Nanoscale Materials was supported by the U.S. Department of Energy, Office of Science, Basic Energy Science, under contract no. DE-AC02-06CH11357. J.W. and Y.Y. were supported by the Ames Laboratory, the U.S. Department of Energy, Office of Science, Basic Energy Sciences, Materials Science and Engineering Division under Contract No. DE-AC02- 07CH11358. XRD and AFM were performed in part at the Chapel Hill Analytical and Nanofabrication Laboratory, CHANL, a member of the North Carolina Research Triangle Nanotechnology Network, RTNN, which is supported by the US National Science Foundation, Grant ECCS-1542015, as part of the National Nanotechnology Coordinated Infrastructure, NNCI. CD measurement was performed at the UNC Macromolecular Interactions Facility, supported by the National Cancer Institute of the US National Institutes of Health under Award No. P30CA016086. We thank Prof. Yizheng Wu of Fudan University for beneficial discussion.

## Author contributions

D.S., W.Z., and H.W. conceived this study and the experiments. K.C. and H.W. performed THz emission measurements. E.V. and L.Y. fabricated the devices. L.Y. characterized the 2D-HMH film with XRD, absorption, AFM, and CD spectra. J.W. and Y.Y. were responsible for the model building of the asymmetric THz emission. Q.Z., Y.L., Y.X., H.Q., R.S., and A.H. assisted in the data discussion. E.V., D.S., and W.Z. wrote the manuscript. H.W., W.Y., W.Z., and D.S. were responsible for the project planning, group managing, and manuscript writing. All authors discussed the results, worked on data analysis and manuscript preparation.

## Competing interests

The authors declare no competing interests.
