## [Peer Review File · Nature Communications]

Reviewers' Comments:

Reviewer #1:

Remarks to the Author:

The authors present their work on THz emission from structures composed of two-dimensional hybrid metal halides (2D-HMH) interfaced with a ferromagnetic metal.

The main finding is the asymmetric intensity for the forward and the backward (reflected from the interface between the sapphire substrate and air) emission direction. Furthermore, the authors find that the asymmetric emission can be controlled by the direction of the applied field and the linear polarization of the laser.

The manuscript is nicely presented. Additionally, the measurements and all the performed control experiments in the supplementary materials give additional credit to the work.

However, the main finding of the paper, meaning the dependence of the THz emission on the linear polarization of the fs laser is difficult to understand. The reason is that the spintronic THz-emission based on the inverse spin Hall effect does not depend on the polarization state of the incident fs laser beam.

The authors assume some kind of modification of the complex dielectric constant by intense light due to Kerr effect at the Rashba state citing the argument from Reference 31. However, in my opinion only the presence of an in-plane photocurrent component parallel to the magnetization could justify the dependence on the linearly polarized light. Can the authors argue on this scenario? Can the authors show in Figure 4 how the E_y component of the emitting radiation is behaving?

Furthermore, the authors claim that the 2D-HMH materials may prove superior to current conventional semiconductor materials for THz applications however there is no proof about this. In addition, I don't see the advantage of this structure with respect to for example NiFe/Pt THz emitter. Are the THz efficiencies comparable with the typical spintronic THz emitters? In conjunction with the last comment, the comparison of bandwidths between NiFe/Pt and NiFe/2D-HMH look similar. If there is such an asymmetry of the pulses and a striking dependence on the polarization that hints to other mechanisms responsible for the radiation why we don't observe any difference in the spectra?

The authors should comment on the role of the SiO₂ capping layer in the THz emission. Can the observed asymmetry have originated from the propagation of pulses through SiO₂?

In Figure 2e the reader observes a saturation of the signal with the thickness of the 2D-HMH layer. Why is there a saturation? The authors comment that the effect hints to an interface origin, but still the THz radiation should suffer from absorption the thicker the sample is, especially for the backward pulse.

In order to exclude any thermal effects on the signal the Fluence dependence of the THz amplitude should be presented.

The authors should explain the experiment in S8. Is it a THz spectroscopy experiment? What was the source of the THz pulse?

The authors should mention the refractive index of sapphire so the reader can validate the time delay between the two pulses.

Figure 3 caption: it is written E_y in the last sentence instead of E_x

Reviewer #2:

Remarks to the Author:

Comments and suggestions

In this work, the authors reported that through combining a kind of energy material of 2D HMH before fabricating ferromagnetic layers so as to observe asymmetric intensity THz radiation from this hybrid materials. Furthermore, they also demonstrate that the linearly polarized THz radiation from this hybrid material can be controlled by the applied external magnetic field direction as well as the pump laser polarization states. For the observed asymmetric intensity THz radiation phenomenon, it is interesting but the possible mechanism is not clear, even the qualitative explanation is not that reasonable to make sense. For the coherent control of the asymmetric spintronic THz emission, the results are not that interesting, since the control method of linearly polarized terahertz waves no matter it is magnetic or optical method.

Therefore, the brightest point of this work lies in the observed phenomenon of asymmetric intensity THz emission from the hybrid materials, but the mechanism explanation is superficial. The work lacks of novelty, and I do not recommend it to be published on Nature Communications.

Besides, I have the following questions.

1. For practical applications, no matter it is in research or industry, the radiation power or efficiency is one of the most important factors to be considered. In the first sentence of the Abstract, the goal of this work is aimed to develop next-generation THz sources, did the authors compare the radiation efficiency with W/CoFeB/Pt trilayer samples or other routing spintronic THz emitters under the same experimental parameters? How about the THz radiation efficiency of this hybrid material?

2. The authors recognized that the observed asymmetric THz radiation strongly correlated with the 2D-HMH layer. They changed the material types. Did they systematically investigate the HMH thickness dependence or not?

3. For the possible mechanism, the authors thought that the generated intense THz pulses inside the sample can modify the complex dielectric constant due to the Kerr effect at the Rashba state. For the first, how did the authors know that the generated THz pulses inside the sample was intense THz? In W/CoFeB/Pt, the excitation laser pulses with 800 nm central wavelength, 1 kHz repetition rate, and 40 fs pulse duration, and the maximum pump energy went up to 4 mJ, and the sample size went up to 3-inch, the generated THz pulse went up to 300 kV/cm (2017, APL, Tobias Kampfrath's group). In this work, the authors gave the pump fluence of 1 mJ/cm², even the radiated THz pulse duration was almost the same as that from W/CoFeB/Pt, but the efficiency, as far as I predict, should be much lower than that from W/CoFeB/Pt. Therefore, this possible mechanism needs further experimental measurement and verification. Even the THz field-induced Kerr effect may be the possible, the authors could use strong-field THz sources from lithium niobate crystal based tilted pulse front technique to verify whether the THz-field induced Kerr effect exist or not. But the authors did not do this.

Detailed response to Reviewers' comments

Reviewer #1:

Overall comment

The authors present their work on THz emission from structures composed of two-dimensional hybrid metal halides (2D-HMH) interfaced with a ferromagnetic metal.

The main finding is the asymmetric intensity for the forward and the backward (reflected from the interface between the sapphire substrate and air) emission direction. Furthermore, the authors find that the asymmetric emission can be controlled by the direction of the applied field and the linear polarization of the laser.

The manuscript is nicely presented. Additionally, the measurements and all the performed control experiments in the supplementary materials give additional credit to the work.

Authors' response: We thank the reviewer for the overall positive comments for our manuscript that our work is “nicely presented”. Particularly the most highlighted finding (i.e., asymmetry THz intensity) has been well-received by this reviewer. In the revised manuscript, we have carefully addressed all his/her comments, and hope the revised manuscript will meet the acceptable level of *Nature Communications*.

Comment 1.1:

However, the main finding of the paper, meaning the dependence of the THz emission on the linear polarization of the fs laser is difficult to understand. The reason is that the spintronic THz-emission based on the inverse spin Hall effect does not depend on the polarization state of the incident fs laser beam.

The authors assume some kind of modification of the complex dielectric constant by intense light due to Kerr effect at the Rashba state citing the argument from Reference 31. However, in my opinion only the presence of an in-plane photocurrent component parallel to the magnetization could justify the dependence on the linearly polarized light. Can the authors argument on this scenario? Can the authors show in Figure 4 how the E_y component of the emitting radiation is behaving?

Authors' response: We thank the reviewer for his/her affirmation of the significance and novelty of our work. It is indeed strikingly surprising to observe asymmetric THz emission depending on the direction of applied magnetic field and linear polarization that have not been reported in either metallic spintronic THz emitters (**Fig. 2b**) or 3D-HMH based THz emitters (see **Fig. S7**).

The scenario that the reviewer proposed has been actually considered as shown in **Fig. 2c**. By reversing the direction of the magnetic field from $+B$ to $-B$, both the THz amplitude and phases along the E_y direction (i.e., in-plane photocurrent parallel to the magnetization) remain unchanged without showing a similar asymmetric behavior as that along the E_x direction. **Res. Fig. 1** shows the detailed measured time trace of the THz pulse along the E_x and E_y direction when $+B$ is applied, respectively. The THz electric field along the E_x direction (blue curve) reverses its phase between the forward and backward emission as one of the distinct features for the spintronic THz pulse generated from 2D-HMH/NiFe heterostructure. However, the subtle THz electric field along the E_y direction (red plot) does not invert its phase. This suggests that the THz emission along the E_y direction belongs to a *different THz source* that is not accounted for the asymmetric spintronic THz signal along the E_x direction.

We also appreciated the suggestion raised by the reviewer about the pump polarization dependence of the E_y component. The THz amplitude along the E_y direction is very weak. It is very challenging to obtain reasonable signals during the pump polarization dependence measurement and thus no clear change can be determined.

Res. Fig. 1. **a**, Measured time trace of the generated THz pulse from the 2D-HMH/NiFe heterostructure at +B, where the forward (peak 1) and backward (peak 2) emission is labeled. **b** and **c**, measured transient THz pulses for the forward and backward THz emission along with two directions, respectively.

Comment 1.2:

Furthermore, the authors claim that the 2D-MHM materials may prove superior to current conventional semiconductor materials for THz applications however there is no proof about this. In addition, I don't see the advantage of this structure with respect to for example NiFe/Pt THz emitter. Are the THz efficiencies comparable with the typical spintronic THz emitters?

Authors' response: We apologized for the confusion in the previous manuscript. On page 3, paragraph 2, we did state that the 2D-HMH-based spintronic THz may prove superior to current semiconductor materials for THz applications in different aspects. In contrast to the semiconductor-based or NiFe/Pt-based THz emitters that require sophisticated deposition approaches and are more susceptible to defects, the advantages of using 2D-HMHs here are mainly due to their *coherent control* capabilities, exceptional *defect-tolerances*, and *solution-processed scalable* device fabrication, but not about the efficiencies.

In the revised manuscript, we have roughly calculated THz efficiencies from the 2D-HMH/NiFe heterostructures as shown in Res. Table 1. Under the same intensity of the fs laser fluence, the obtained nominal THz electric field at the 2D-HMH/NiFe interface ($\sim 22 \times 10^{-6}$ a.u.) is roughly **10 times** smaller compared with that in a typical NiFe/Pt spintronic THz emitter measured with the same experimental conditions ($\sim 230 \times 10^{-6}$ a.u.). Thus, we estimated the THz efficiency in the 2D-HMH/NiFe heterostructure is roughly **10 times** smaller. Derived from the reported THz electric field values in the typical NiFe/Pt device (~ 100 kV/cm), the THz field from the 2D-HMH/NiFe device is around 10 kV/cm.

In the revised manuscript (page 3, paragraph 2, line 12), this sentence has been rephrased as: “*The improved stability and scalable thin film process of the reduced-dimensional HMH materials*

enable us to meet the emerging needs for low-cost THz sources with coherent control capabilities”.

Comment 1.3:

In conjunction with the last comment, the comparison of bandwidths between NiFe/Pt and NiFe/2D-HMH look similar. If there is such an asymmetry of the pulses and a striking dependence on the polarization that hints to other mechanisms responsible for the radiation, why we don't observe any difference in the spectra?

Authors' response: We thank the reviewer's note about the bandwidth and shape of the THz spectra. We have carefully discussed this concern on page 6, paragraph 2, line 4. We stated: “*While the 1st and 2nd peak groups exhibit different intensity, the B(+)/B(-) traces are roughly aligned with each other suggesting an equal intensity of overall THz signals stemming from the same amount of superdiffusive spin current in the 2D-HMH film at opposite magnetic fields.*” Whereas the asymmetric THz modulates the distribution of THz pulses toward two opposite directions, the overall THz intensity is still the same. The obtained THz spectra at two magnetic fields contain a sum of THz amplitudes from both forward and backward THz emissions. Thus, no obvious difference in the spectra domain can be observed.

Res. Fig. 2. Comparison of the THz spectra in the NiFe/Pt and 2D-HMH/NiFe device, respectively.

Moreover, the THz spectra obtained from the 2D-HMH/NiFe device indeed exhibit a distinct frequency domain in contrast to that in the NiFe/Pt device. Please see **Res. Fig. 2** and **S.I. Fig. S4**. We found the maximum THz frequency from the NiFe/Pt device is located at $f_c \approx 0.4$ THz, whereas the one from the 2D-HMH/Pt device is at **0.7 THz**. The -10dB point from the NiFe/Pt device is located at **1.5 THz** while the THz bandwidth from the 2D-HMH/NiFe device is broader (up to **2.5 THz**).

Comment 1.4:

The authors should comment on the role of the SiO₂ capping layer in the THz emission. Can the observed asymmetry have originated from the propagation of pulses through SiO₂?

Authors' response: We thank the reviewer for this constructive comment that helps us to revisit the role of SiO₂ for this effect. It was reported that the Rashba interface can be formed when the ferromagnet layer is interfaced with SiO₂ or Sapphire substrate which can contribute to the generation of the THz emission.

To fully separate the contribution from the possible Rashba interface formed at the SiO₂ or Sapphire substrate [e.g., Chen et al., Nat. Commun. 9, 2569 (2018), Luo et al., Phys. Rev. Applied 11, 064021 (2019), etc.], we have performed a series of spintronic THz experiments in multiple control samples, including Sapphire/SiO₂/NiFe, Sapphire/NiFe/SiO₂, Sapphire/NiFe/Pt, and Sapphire/Pt/NiFe, Sapphire/2D-HMH/NiFe, and Sapphire/3D-HMH/NiFe devices. We measured the THz emission (a.u.) in each device under the same conditions (field strength, laser intensity, sample position, etc.). Following the literature reports, we assumed the formation of the interfacial Rashba/SOC interface between the Py (NiFe) and the Sapphire/SiO₂ substrate (marked as ||) that is responsible for the measured THz electric field. By systematically changing the device stacking configuration, the role of the Rashba effect at the SiO₂ or sapphire interface can be well-separated, from which the actual THz emission from the 2D-HMH/NiFe interface can be derived (see **Res. Table 1** below).

We found the THz emission from the potential Rashba interface formed at the Sapphire||Py interface is indeed strong. Its amplitude (84×10^{-6} a.u.) is comparable to that from the Py||Pt interface (146×10^{-6} a.u.). However it is noteworthy in our HMH-based THz configuration, there is no direct contact between Py and Sapphire. Consequently, this interface will not be responsible for the measure THz emission in the 2D-HMH/Py device.

The THz emission from Py||SiO₂ interface is roughly three times weaker (6×10^{-6} a.u.) than that from 2D-HMH||Py interface (22×10^{-6} a.u.). This confirms that the possible Rashba/SOC interface between Py and SiO₂ (if it exists) is not the dominant THz source for the observed signals.

Res. Table 1. Obtained THz intensities in different device structures at the applied positive magnetic field, +B. The THz signal originating from 2D-HMH materials has been calculated. Note: the reversed interface yields opposite THz signal, as confirmed by the reversed THz polarity between the NiFe/Pt and Pt/NiFe device.

Devices (: normal interface; : Rashba/SOC interface)	THz (10^{-6}) from the left interface of Py	THz (10^{-6}) from the right interface of Py	Observed THz field (10^{-6})
Sapphire SiO ₂ Py	+6	0	+6
Sapphire Py SiO ₂	-84	-6	-90

Sapphire 2D-HMH Py SiO ₂	+22	-6	+16
Sapphire 3D-HMH Py SiO ₂	+10	-6	+4
Sapphire Py Pt	-84	-146	-230

Although the THz intensity from the SiO₂/Py interface does not play a significant role in the observed asymmetric THz emission from the 2D-HMH/Py heterostructures, we indeed observed an *asymmetric THz emission* as a function of the relative angle (θ) between pump linear polarization axis with respect to the magnetic field, as shown in **Res. Fig. 3**. However, the change of THz intensity exhibits a different angular dependence ($\propto \sin(2\theta)$) of which the phase is shifted by **90 degrees** in contrast that in the 2D-HMH/Py device ($\propto \cos(2\theta)$). Therefore, the THz emission from SiO₂/Py interface alone cannot explain the observed angular dependence in our sample.

Res. Fig. 3. Pump polarization dependence of asymmetric THz radiation in Sapphire/SiO₂/Py sample. **a**, 2D contour plot of the electric field of THz radiation along the x-direction as a function of time and polarization angle (θ), exhibiting strong linear polarization-modulated THz intensity in both two peak groups. **b** and **c** summarize the opposite changes of the polarization-dependent THz intensity in Peak 1 and Peak 2 with the switching of the magnetic field, respectively.

To further validate that the observed asymmetric THz emission was mainly attributed to the 2D-HMH/Py interface, we have fabricated a series of reduced-dimensional HMH-based spintronic THz devices (i.e., (BA)₂(MA)_{n-1}Pb_nI_{3n+1}/Py/SiO₂, MA= methylammonium, BA= butylammonium) while maintaining the Py/SiO₂ interface unchanged. **Res. Fig. 4** shows the obtained magnetic-field-switchable THz emission in thin-film reduced-dimensional HMH-based THz emitters having quantum layers $n=1, 2,$ and $4,$ respectively. In these devices, the number of

inorganic PbI framework increases thus substantially changes the Rashba interface between RD-HMH and Py while the Py/SiO₂ interface remains the same. If the Py/SiO₂ interface dominates the THz signal, the asymmetric feature would be also unchanged. By increasing the quantum layer from n=1 to n=4, we found that the difference of THz intensity between the positive and negative magnetic fields increases with the increasing quantum layer. This suggests the key role of the quantum well effect for the asymmetric THz emission and their correlation with the layer-dependent Rashba state [Yin et al., Chem. Mater. 30, 8538 (2018)], which is confirmed to be separated from the Py/SiO₂ interface.

Res. Fig. 4. Observations of THz emission in $(\text{BA})_2(\text{MA})_{n-1}\text{Pb}_n\text{I}_{3n+1}/\text{NiFe}$ devices with the quantum number, $n=1, 2,$ and $4,$ respectively.

Comment 1.5:

In Figure 2e the reader observes a saturation of the signal with the thickness of the 2D-HMH layer. Why is there a saturation? The authors comment that the effect hints to an interface origin, but still the THz radiation should suffer from absorption the thicker the sample is, especially for the backward pulse.

Authors' response: The saturation of the THz intensity has been attributed to the quality of the interface formed between the 2D-HMH and NiFe layer. Different from the MBE-growth of the conventional semiconductor layer, the 2D-HMH films were spin-coated on the sapphire substrate in the glove box using a recipe particularly designed for HMH-based photovoltaic applications. This receipt provides the optimized thickness of the 2D-HMH layer above 100nm. The prepared polycrystalline 2D-HMH thin films exhibit the optimized morphology and smooth surface due to

the synergistic effects of the chemical reactions and crystalline domain formation during the solution process that have been reported elsewhere [e.g., Hu et al., *Advanced Materials* 30, 1802041 (2018)]. The smoothness of the HMH-layer influences the quality of the interface and the deposited thin NiFe layer (~5 nm), yielding the optimized THz device performance at higher thickness.

The THz absorption for the thin layer of the HMH-based device has been studied in **S.I. Fig. S8**. At the current 2D-HMH thickness ranging between 50nm-150nm, no clear THz absorption was observed compared with that in the reference sample.

Comment 1.6:

To exclude any thermal effects on the signal the Fluence dependence of the THz amplitude should be presented.

Authors' response: We thank the reviewer for his/her suggestion. In the revised manuscript, fluence dependence of the THz intensity has been added to the S.I. section, as shown in **Res. Fig. 5** below. It shows that the THz electric field scales nearly linearly at low laser pump fluences (< 0.8 mJ/cm²), following by a saturating behavior at higher laser fluences. This pronounced sub-linear fluence dependence is consistent with that in Co/Pt THz devices [Huisman et al, *Nat. Nano.* 11, 455 (2016)]. It excludes possible heat-driven spin generation in the NiFe layer induced by the laser pulse and resulting THz emission since this scenario would result in a superlinear fluence dependence of the THz emission. However, it is possible that the high laser fluences may induce an unavoidable sample degradation because of the laser absorption in the metallic NiFe layer. This may lead to a change of the THz conductivity in the HMHs owing to their low-thermal conductivities [e.g., 0.5 W·K⁻¹·m⁻¹ in CH₃NH₃PbI₃, *J. Phys. Chem. Lett.* 5, 2488 (2014)] and decreased magnetization of the NiFe layer. Both would be accounted for the saturation behavior of the THz signal at higher laser fluences. To trade-off between the THz intensity and sample degradation induced by a strong laser pulse, for all our THz measurements, the pump beam with a pump fluence of 1.0 mJ/cm² is used to excite the devices in the normal incidence.

Res. Fig. 5. Fluence dependence of the peak-to-peak THz amplitude obtained from the 2D-HMH/NiFe heterostructure.

Comment 1.7:

The authors should explain the experiment in S8. Is it a THz spectroscopy experiment? What was the source of the THz pulse?

Authors' response: We thank the reviewer for this note. **Fig. S8** is the obtained THz transmission through the 2D-HMH/NiFe heterostructure. The THz pulses are generated from optical rectification of 800 nm pulses from Ti: Sapphire regenerative amplifier (Spectra-Physics Spitfire Ace) in a 0.3 mm-thick GaP crystal. . The transmission of the THz through the devices was measured using the same detection scheme as in the THz emission experiment. We have added these technical details to the **S.I. Fig. S8** of the S.I.

We found that there is no detectable difference for the THz transmission and absorption along the forward (+z) and backward (-z) directions. Thus, we confirmed that the observed asymmetric THz emission is not caused by the possible birefringence of the heterostructure or the substrate.

Comment 1.8:

The authors should mention the refractive index of sapphire so the reader can validate the time delay between the two pulses.

Authors' response: We thank the reviewer's suggestion. In the revised manuscript, we have added the refractive index of sapphire in the method section. The thickness of the used sapphire substrate (C-plane (0001), MTI Corp.) is 0.5 mm. The reported dielectric constant for this substrate is 11.58 along the c axis. The refractive index (n) is ~ 3.4 . The delayed pulse peak is reflected at the interface between the sapphire substrate and the air bypassing the substrate twice as indicated in **Fig. 1b**. The total delayed time Δt is $(0.5\text{mm} \times 2)/(c/n) \approx 11\text{ps}$, consistent with our observations.

Comment 1.9:

Figure 3 caption: it is written E_y in the last sentence instead of E_x .

Authors' response: We thank the reviewer to point this out. In the revised manuscript, we have fixed this mistake.

Reviewer #2:

Overall comment

In this work, the authors reported that through combing a kind of energy material of 2D HMH before fabricating ferromagnetic layers so as to observe asymmetric intensity THz radiation from this hybrid materials. Furthermore, they also demonstrate that the linearly polarized THz radiation from this hybrid material can be controlled by the applied external magnetic field direction as well as the pump laser polarization states. For the observed asymmetric intensity THz radiation phenomenon, it is interesting but the possible mechanism is not clear, even the qualitatively explanation is not that reasonable to make sense. For the coherent control of the asymmetric spintronic THz emission, the results are not that interesting, since the control method of linearly polarized terahertz waves no matter it is magnetic or optical method.

Therefore, the brightest point of this work lies in the observed phenomenon of asymmetric

intensity THz emission from the hybrid materials, but the mechanism explanation is superficial. The work lacks of novelty, and I do not recommend it to be published on Nature Communications.

Authors' response: We thank the reviewer's effort in reviewing our manuscript and for the recognition of our highlighted points of this work about the observation of asymmetric THz emission from the 2D-HMH/NiFe heterostructures. The major concern of this reviewer is about the novelty of our work. Below we emphasize again the high merits of our work:

Our work presents the **first report of asymmetric spintronic-THz emission in 2D-HMHs/ferromagnet heterostructures at room temperature with coherent control capabilities.** It opens opportunities for directional control of THz emission, a new control scheme that is distinct from polarization controls of the THz emission as reported before. We show that:

- (1) The emitted asymmetric THz electric fields can be generated by an ultrafast, transient spin current pulse from the thin, adjacent ferromagnet layer (i.e., NiFe) followed by spin-to-charge conversion at femtosecond timescales in the solution-processed 2D-HMH materials.
- (2) The emitted THz radiation is directional. The forward and backward emission intensity can be coherently controllable by incident light polarization and magnetic field. This is in sharp contrast to that in the prototypical metallic heterostructures (e.g., NiFe/Pt) and 3D-HMH materials where the emitted THz field intensity is mostly independent of the magnetization direction.
- (3) A large percentage of the THz amplitude, up to 30-40%, can be modulated by the pump linear polarization, demonstrating the realization of coherent control of THz amplitude in the hybrid spintronic-THz emitter.

Our work shows that 2D-HMHs would be desirable material candidates for spintronic-THz generation and manipulation. It offers new routes towards interconversion between photons, charges, and spin states using scalable, printable, solution-processed hybrid compounds with contactless coherent control capabilities utilizing spintronic toolkits. The excellent optoelectronic, spin, and carrier transport properties of HMH materials are also known to exhibit exceptional defect tolerance. Formation of shallow trapped states when these materials are processed using cruder fabrication methods than conventional electronic materials contribute to their fault tolerance, which is a critical *material-by-design* requirement for spin-based applications at room temperature. This suggests that the 2D-HMH materials may prove superior to the current conventional semiconductor materials for THz applications, which require sophisticated deposition approaches that are more susceptible to defects. (i.e., low defect tolerance).

For THz technologies, the potential broadband THz emission generated by spintronic THz emitters outperforms the emitters such as ZnTe (110) crystals in terms of scalability and cost. The solution-based, on-chip fabrication of HMH/FM heterostructures is not only well compatible with Si-based technology but also fits with other amorphous/crystalline substrates (e.g., <100 μm fused silica) for tailoring the THz emission via patterning design. The linearly polarized electric field direction of the THz emission strongly depends on the spin states of the ferromagnet layers, offering 'sensitive and active' control of the asymmetric THz emission.

For materials research, HMH materials exhibit a vast and unexplored chemical 'universe' which includes perovskite, Ruddlesden-Popper, and Dion-Jacobson structures, through the combination

of various metal halides and organic molecular cations. Thus, our research will launch a promising testbed for designing a wide variety of low-dimensional HMM materials for future solution-based spintronic and spin-optoelectronic applications, whereas similar materials development in the class of conventional semiconductors has been largely encumbered. The further improved stability and scalable thin film process of the reduced-dimensional HMM materials enable us to meet the emerging needs for low-cost THz sources.

Comment 2.1:

For practical applications, no matter it is in research or industry, the radiation power or efficiency is one of the most important factors to be considered. In the first sentence of the Abstract, the goal of this work is aimed to develop next-generation THz sources, did the authors compare the radiation efficiency with W/CoFeB/Pt trilayer samples or other routing spintronic THz emitters under the same experimental parameters? How about the THz radiation efficiency of this hybrid material?

Authors' response: We apologize that the confusion in the abstract of the previous manuscript. We aim to constitute next-generation spintronic THz source using solution-process HMM materials by taking advantage of their *coherent control* capabilities, exceptional *defect-tolerances*, and *solution-processed scalable* device fabrication, but not about the efficiencies. We have rephrased the abstract in the revised manuscript. For the detailed estimation of THz efficiencies in our devices, please see our response to comment #1.2

We want to point out that the THz efficiency is **not the focus** of our current work, although we believe that the THz efficiency in the 2D-HMM-based THz devices can be further boosted by optimizing the device stacking configuration [Seifert et al., Nat. Nano. 10, 483 (2016)] or through the HMM material search. The family of HMM materials exhibits a vast and unexplored chemical 'universe' through the combination of various metal halides and organic molecular cations. It has been reported that the Rashba parameters (α_R) in one of 2D-HMMs or 3D-HMM materials [Zhai et al., Sci. Adv. 28, e1700704 (2017); Niesner et al., Phys. Rev. Lett. 117, 126401 (2016)] can be up to 1.6 or even 11 eV·Å that could be potentially used for very efficient spintronic THz emitters comparable with the metallic ones. We hope we could discuss and optimize the device performance for future publications.

Comment 2.2:

The authors recognized that the observed asymmetric THz radiation strongly correlated with the 2D-HMM layer. They changed the material types. Did they systematically investigate the HMM thickness dependence or not?

Authors' response: We have included the thickness-dependence results in Fig. 2e. Please also see our responses to comments #1.4 and #1.5.

Comment 2.3:

For the possible mechanism, the authors thought that the generated intense THz pulses inside the sample can modify the complex dielectric constant due to the Kerr effect at the Rashba state. For the first, how did the authors know that the generated THz pulses inside the sample was intense THz? In W/CoFeB/Pt, the excitation laser pulses with 800 nm central wavelength, 1 kHz repetition rate, and 40 fs pulse duration, and the maximum pump energy went up to 4 mJ, and the sample size went up to 3-inch, the generated THz pulse went up to 300 kV/cm (2017, APL, Tobias

Kampfrath's group). In this work, the authors gave the pump fluence of 1 mJ/cm², even the radiated THz pulse duration was almost the same as that from W/CoFeB/Pt, but the efficiency, as far as I predict, should be much lower than that from W/CoFeB/Pt. Therefore, this possible mechanism needs further experimental measurement and verification. Even the THz field-induced Kerr effect may be the possible, the authors could use strong-field THz sources from lithium niobate crystal based tilted pulse front technique to verify whether the THz-field induced Kerr effect exist or not. But the authors did not do this.

Authors' response: We thank the reviewer for pointing this out. The reviewer may misunderstand our previous tentative model in the previous manuscript. We agree that the generated THz pulse (~10 kV/cm) is weaker than that in optimized metallic spintronic THz emitters and can alter material properties. However, the incident pump laser (not the generated THz pulse) is intense and can potentially induce the Kerr effect that we proposed. Therefore, the THz emission can depend on the angle between the linear polarization of the optical light and the magnetization of the NiFe layer, as illustrated in **Fig. 4**.

Moreover, since one of the major concerns from reviewer #2 is the theoretical interpretation, in the revised manuscript we developed a new theoretical model to explain the observed asymmetry THz radiation between two magnetic fields. Please see page 10 and **Fig. 5** for a detailed discussion.

Last, we particularly appreciate reviewer #2's challenge about the novelty of our work and the theoretical model. Based on these insightful comments, our extensive revision has clarified the novelty and presented a new theoretical model, which has greatly improved the manuscript. We hope that the reviewer could share our enthusiasm for the beauty of *experimental techniques*, *remarkable device performance*, and *underlying new physics* that we reveal here, which are at present *Terra incognita*. We hope we could have this reviewer's support for the publication of our work in *Nature Communications*.

Reviewers' Comments:

Reviewer #1:

Remarks to the Author:

The authors have successfully answered all the points of criticism by performing additional experiments and by proposing a new theoretical model.

The manuscript is significantly improved, I feel that the work can be now published.

I only have two small remarks:

-How much is the intensity decreased by using a Cu interlayer in Fig. 2f?

-I suggest the authors to make clearer to the reader the origin of the electric dipole along the Z-axis.

Technical:

-In Figure 5 the Pdipole is misspelt.

Reviewer #2:

Remarks to the Author:

The authors have answered my concerns and questions, and now I agree to accept this article

Detailed response to Reviewers' comments

Reviewer #1:

Overall comment

The authors have successfully answered all the points of criticism by performing additional experiments and by proposing a new theoretical model. The manuscript is significantly improved, I feel that the work can be now published.

Authors' response: We thank the reviewer's positive feedback on our manuscript, and for recommending publication of our work in Nature Communications. The comments have helped us to significantly improve the quality of our presentation.

Comment 1.1:

How much is the intensity decreased by using a Cu interlayer in Fig. 2f?

Authors' response: We thank the reviewer for this comment. We have roughly calculated the intensity of the THz emission (forward direction) from the 2D-HMH/Cu(10nm)/NiFe/SiO₂ sample, which is about 12×10^{-6} arb. units after subtracting the contribution from the NiFe/SiO₂ interface (as shown in Table S1). This value is decreased compared to that in a typical 2D-HMH/NiFe/SiO₂ sample without having the Cu interlayer (16×10^{-6} arb. units). The reduction of THz radiation might be caused by (i) an extra attenuation when the THz radiation passes the additional Cu metallic layer (10 nm) during the THz measurement; (2) slight reduction of the injected transient spin current from NiFe to the 2D-HMH by passing the Cu interlayer; and (3) larger conductivity mismatch at the 2D-HMH/Cu interface compared to that at the 2D-HMH/NiFe interface.

Comment 1.2:

I suggest the authors to make clearer to the reader the origin of the electric dipole along the Z-axis.

Authors' response: We thank the reviewer for this comment. On page 11, we have elaborated the origin of the electric dipole generated by the laser pulse.

Comment 1.3:

In Figure 5 the Pdipole is misspelt.

Authors' response: Thanks for catching this. In the revised manuscript, Figure 5 has been updated.

Reviewer #2:

Overall comment

The authors have answered my concerns and questions, and now I agree to accept this article.

Authors' response: We thank the reviewer's positive feedback on our manuscript, and for recommending publication of our work in Nature Communications.